# Triaging and referring in adjacent general and emergency departments (the TRIAGE trial): A cluster randomised controlled trial

Stefan Morreel [1]*, Hilde Philips[1], Diana De Graeve[2], Koenraad G. Monsieurs[3,4], Jarl K. Kampen[5], Jasmine Meysman[2], Eva Lefevre[2], Veronique Verhoeven[1]

1 Department of Family and Population Health, University of Antwerp, Antwerp, Belgium, 2 Department of Economics, University of Antwerp, Antwerp, Belgium, 3 Department ASTARC, University of Antwerp, Antwerp, Belgium, 4 Emergency Department, Antwerp University Hospital, Antwerp, Belgium, 5 Department of Epidemiology and Medical Statistics, Antwerp University Hospital, Antwerp, Belgium

* stefan.morreel@uantwerpen.be

**Data Availability Statement:** The studied data is available to researchers worldwide after following the application procedures of iCAREdata (see

## Abstract

### Objectives

To determine whether a new triage system safely diverts a proportion of emergency department (ED) patients to a general practitioner cooperative (GPC).

### Methods

Unblinded randomised controlled trial with weekends serving as clusters (three intervention clusters for each control). The intervention was triage by a nurse using a new extension to the Manchester Triage System assigning low-risk patients to the GPC. During intervention weekends, patients were encouraged to follow this assignment; it was not communicated during control weekends (all patients remained at the ED). The primary outcome was the proportion of patients assigned to and handled by the GPC during intervention weekends. The trial was randomised for the secondary outcome: the proportion of patients assigned to the GPC. Additional outcomes were association of these outcomes with possible confounders (study tool parameters, nurse, and patient characteristics), proportion of patients referred back to the ED by the GPC, hospitalisations, and performance of the study tool to detect primary care patients (the opinion of the treating physician was the gold standard).

### Results

In the intervention group, 838/6294 patients (13.3%, 95% CI 12.5 to 14.2) were assigned to the GPC, in the control group this was 431/1744 (24.7%, 95% CI 22.7 to 26.8). In total, 599/6294 patients (9.5%, 95% CI 8.8 to 10.3) experienced the primary outcome which was influenced by the reason for encounter, age, and the nurse. 24/599 patients (4.0%, 95% CI 2.7 to 5.9) were referred back to the ED, three were hospitalised. Positive and negative predictive values of the studied tool during intervention weekends were 0.96 (95%CI 0.94 to 0.97) and 0.60 (95% CI 0.58 to 0.62). Out of the patients assigned to the GPC, 2.4% (95% CI 1.7 to 3.4) were hospitalised.

icaredata.eu). Given the privacy policy of the iCAREdata database, the authors are not allowed to share the used data as supporting information or in a public repository. Sharing this database would potentially harm the privacy of the included patients. The Belgian Data Protection Authority does not allow the authors to share the raw data. The authors are, however, able to deliver a selection of variables and the outputs of their statistical software upon reasonable request. Such a request should be directed towards the authors or icaredata@uantwerpen.be.

**Funding:** All authors received a grant (number T000718N) from Fonds Wetenschappelijk onderzoek (https://www.fwo.be/) for this project, covering the personnel and working costs; payment was made to their institutions. The funders had no role in study design, data collection and analysis, decision to publish, or preparation of the manuscript.

**Competing interests:** I have read the journal's policy and the authors of this manuscript have the following competing interests: The second author is a general practitioner working in the surroundings of the study site, and as such, he performed on call shifts at the study site and treated some of the studied patients. Due to the anonymity of the studied data, the exact number of study patients seen by him cannot be determined, but it was definitely below ten. He is also a board member of the studied general practice cooperative receiving meeting fees. Author HP is coordinator of the iCAREdata project (database used for this study). She had an appointment at the University of Antwerp for this project until September 2020. The authors declare no other relationships or activities that could appear to have influenced the submitted work.

## Conclusions

ED nurses using a new tool safely diverted 9.5% of the included patients to primary care.

## Trial registration

ClinicalTrials.gov Identifier: NCT03793972

## Introduction

In many countries, out-of-hours (OOH) primary care is increasingly organised in General Practitioner Cooperatives (GPCs), and simultaneously, emergency care is provided by emergency departments (EDs) in hospitals. Although there is no clear definition of 'appropriate' or 'inappropriate' use of the ED, several authors reported that many medical problems presented at the ED could be managed in a primary care setting [1–4]. In the United States, primary care office visits for acute care dropped sharply in 2002–15, while ED visits increased modestly [5]. In Belgium, an ED has the legal obligation to assess and to treat all patients with an emergency medical condition regardless of an individual's ability to pay, which is very similar to the Emergency Medical Treatment & Labor Act in the United States [6]. Patients choose a service based on previous experiences, ease of access, explanation by the doctor about the illness and treatment, the anticipated waiting time, their relationship with their general practitioner (GP), and the perceived nature of the complaint [7]. Diverting patients in emergency departments to primary care services helps patients to make this choice, but little is known about its safety and effectivity [8–10]. Both patients and physicians in Belgium are in favour of co-locating these services [11]. Improved access to OOH primary care was associated with increased primary care utilisation but did not necessarily lead to a decrease of workload at the ED [9]. At the time of the current study, Belgian GPCs were only open during weekends and bank holidays.

Triage is defined as the sorting out and classification of patients or casualties to determine priority of need (urgency classification) and proper place of treatment (in the current study assignment to ED or GPC) [12]. Before this trial, almost all EDs in Belgium used nurse triage to determine priority and place of treatment within the hospital but diverging patients to a GPC was only done in experimental settings [13]. The Manchester Triage System (MTS) is one of the few triage systems with a moderate to good validity, it is used worldwide [14]. Three, non-randomised trials about the MTS and diversion to primary care revealed promising results but did not allow for definitive conclusions about safety and effectiveness because of their small sample size and focus on specific groups of patients such as children [15–17]. An awareness-raising campaign was conducted as a pilot for the current study in order to collect baseline data, to assess the feasibility of local cooperation and to estimate the needed sample size [18].

The objective of the current study was to determine whether a new triage system safely diverts a proportion of emergency department (ED) patients to a general practitioner cooperative (GPC). The trial design is a clustered randomised trial with weekends and bank holidays (from here out we refer to weekends and bank holidays as weekends) serving as units of randomisation and patients as units of analysis. Individual randomisation was not desirable because the triage process is by nature applied to a longer period of at least one working shift. A process and economic analysis of the present trial will be published separately.

## Materials and methods

### Study design, setting and participants

Single centre randomised controlled trial from 01/03/2019 to 30/12/2019. Weekends (7.00 PM Friday to 6.00 AM Monday) served as units of randomisation (approximately 200 participants each) and patients as units of analysis. A preparation period of two months for adapting the software, testing procedures, and training the staff was followed by the actual study (01/03/2019 to 30/12/2019).

This study was performed in the ED of a general hospital staffed by approximately 25 nurses and 10 physicians handling 36 743 contacts in 2018. The adjacent GPC, which is open during OOH care, covers a population of 145 000 inhabitants and handled 10 586 consultations in 2018. All 110 GPs working in the area covered by this GPC are obliged to work there approximately one shift per month. The surrounding area is ethnically diverse with a mix of middle-income and socially deprived neighbourhoods. The Belgian healthcare system is organised into primary, secondary, and tertiary care, with open access for patients to all levels. It is mainly organised as a fee-for-service system.

All patients with a national insurance number triaged by a nurse at the ED were included. Patients arriving at the ED by an ambulance staffed with a doctor or nurse, patients already admitted to the hospital, and patients referred to the ED by a doctor were excluded because they already underwent triage. See **S1 File.** for the entire study protocol and **S2 File.**

### Materials

The MTS (version 3.6) is a tool for prioritisation in the ED. When using the MTS, the nurse chooses one out of 53 presentational flowcharts each concerning a reason for encounter (e.g., abdominal pain in children). A flowchart consists of a list of discriminators (e.g., mild pain), the presence of which has to be checked in a top-down order. Each discriminator is linked to an urgency category ranging from one (immediate care necessary) to five (non-urgent).

For the current study, an extended version of the MTS (eMTS) was created. First, a questionnaire was distributed to a working group consisting of three GPs, two ED-nurses, and two ED physicians. Next, the working group drafted the eMTS during five consensus meetings. The aim of this tool is to identify low-urgency patient eligible for primary care. Due to legal concerns, only patients in urgency categories four and five were allowed to be assigned to the GPC but not all of them are eligible for primary care as some might need hospital care (radiology, complex interventions, hospitalisation. For example, a patient with a deformed joint with mild pain probably needs hospital care (radiology) but has a low urgency category. The working group chose to allow assignment to the GPC when an expected 90% of the GPs would be able to safely help the patient. Wounds requiring sutures for example were assigned to the ED. Babies less than three months old were always assigned to the ED, as some of the GPs might not have enough paediatric experience.

The MTS flowcharts for self-harm, collapse, abused or neglected child, apparently drunk, major incident, behaving strangely, and unwell newborn were not extended. In 20 flowcharts, additional discriminators were created which had to be assessed whenever the urgency category was four or five. Presence of one of these additional discriminators means an assignment to the ED (see Fig 1 for an example). In 26 flowcharts, the only added discriminator was "GP Risk", defined as an unspecified risk to assign the patient to the GPC according to the opinion of the triaging nurse, or because of age less than three months. The eMTS was integrated into a computer decision support system (E.care ED 4.1) that showed "assign to GPC" when

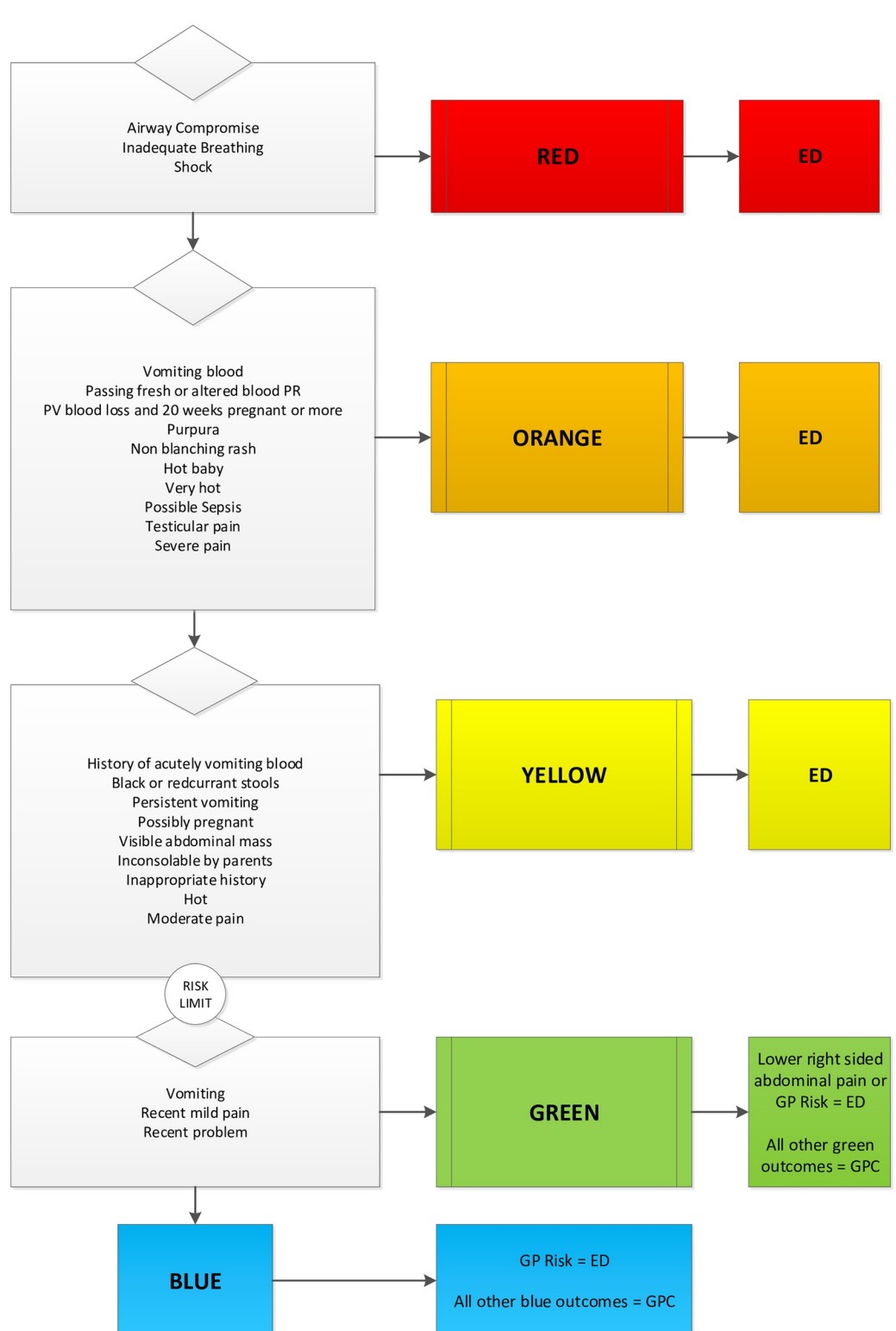

## Abdominal pain in children (2)

If the patient is under 28 days, the Unwell newborn chart should be used

Airway Compromise
Inadequate Breathing
Shock

**RED** → ED

Vomiting blood
Passing fresh or altered blood PR
PV blood loss and 20 weeks pregnant or more
Purpura
Non blanching rash
Hot baby
Very hot
Possible Sepsis
Testicular pain
Severe pain

**ORANGE** → ED

History of acutely vomiting blood
Black or redcurrant stools
Persistent vomiting
Possibly pregnant
Visible abdominal mass
Inconsolable by parents
Inappropriate history
Hot
Moderate pain

**YELLOW** → ED

RISK LIMIT

Vomiting
Recent mild pain
Recent problem

**GREEN**

Lower right sided abdominal pain or GP Risk = ED

All other green outcomes = GPC

**BLUE**

GP Risk = ED

All other blue outcomes = GPC

**Fig 1. Example of a Manchester triage system presentational flowchart with the studied extension.** GP: General Practitioner. PV: Per Vaginam. Image based on Emergency Triage: Mackway-Jones K, Marsden J, Windle J, Manchester Triage Group. Emergency triage. Third edition. Ed, 2014, ISBN 9781118299067 p. 66 with kind permission.

appropriate. The nurses were allowed to overrule the result of this automated eMTS assignment. The eMTS is available upon request.

## Intervention

All patients presenting at the ED were triaged by an experienced nurse using the eMTS, resulting in an urgency level (one to five) and an assignment (to GPC or to ED). The study was only conducted during OOH care as there were no centralised primary care services available during working hours. In a Belgian observational study, only 1.7% of the GPC patients were in need of urgent hospital care so patients presenting at the GPC were not triaged in the current study [19]. During control weekends the assignment was not communicated to the patients, they all remained at the ED. During intervention weekends patients were encouraged to comply to the assignment but were allowed to refuse it. Patients were informed in the ED about the study using flyers, posters, and a presentation broadcasted on a screen. During intervention weekends, this presentation contained additional information in five languages about the possibility of being assigned to the GPC. All nurses followed a twelve-hour training on using the eMTS, patient communication skills, and the study protocol. In one-hour sessions, the researchers informed emergency physicians (participation 80%) and GPs (participation 33%) about the study.

## Outcomes

The primary outcome was the proportion of patients assigned to the GPC and handled by the GPC during intervention weekends. The secondary outcome was the proportion of patients assigned to the GPC during intervention and control weekends. Additional outcomes were the proportion of patients who did not comply to the assignment; the association between the primary and secondary outcomes and possible confounders (study tool parameters, nurse, patient characteristics, and timing of presentation); proportion of patients within the primary outcome referred back to the ED; admissions to the study hospital; and performance of the eMTS as an instrument to detect patients primary care patients (i.e., a low risk of hospital care). The exploration of the primary outcome after the trial ended was the only outcome not pre-specified, it was added to exclude a Hawthorne effect (the changes in behaviour found are caused by the impact of being studied, not by the intervention) [20].

## Sample size

The impact size of the determinants of the primary and secondary outcomes were unknown prior to this study but were expected to be in the order of 10–20%. Therefore, based on known volumes of inflow of patients, two weekends (one intervention and one control) would have been sufficient to provide empirical evidence of a statistically significant shift of patients from the ED to the GPC. However, multivariate analyses of the primary and secondary outcome, the additional outcomes (safety), monitoring of serious adverse events and assessment of a possible learning curve required data collection over a longer period of time. Consequently, a convenience sample of 48 weekends months was selected.

## Randomisation

Because the primary outcome did not apply to the control group (the study needed randomisation for the secondary outcome and for future financial and process analysis), and because more data on the intervention weekends were needed to assess the additional outcomes, a ratio of three intervention weekends for each control was chosen. The trial intentionally started with two intervention weekends. The authors used an algorithm in Microsoft Excel 2016 to generate random allocation stratified for bank and school holidays, while no more than five consecutive intervention weekends were allowed. The head nurse and one assistant were aware of the randomisation. The ED staff were informed a few hours before their shift. The GPC staff were not informed but could find out during their shift. Patients were not blinded.

## Data collection

The following patient characteristics were collected: sex; birthyear; postal code; socioeconomic status (reimbursement status of the Belgian health insurance: increased reimbursement or not); type of admission to the ED (walk-in, arrived by ambulance, or already admitted to hospital); origin (self-referral, referral by GP, referral by specialist); ED physician's post hoc opinion on assignment (to GP or to ED); GP referral back to the ED; admission to the study hospital, and triaging nurse (anonymous identifier ranging from one to 22). After triage, the following study tool parameters were collected: MTS flowchart (52 flowcharts reported in 15 categories); eMTS discriminator; MTS urgency level (one to five), and assignment (ED or GPC). The timing of presentation was registered both at the ED and, when applicable, at the GPC. It had three characteristics: weekend identifier, time period (day, evening, or night), and subjective crowding at the ED (quiet, normal, and busy). Except the subjective crowding, all variables were part of the routine medical records.

In order to calculate the complete number of exclusions, the number of patients without a national insurance number was extracted from the ED's software. All other data were collected using iCAREdata, a database for OOH care [21, 22] iCAREdata links data from the ED and the GPC to each other using the pseudonymised national insurance number.

The studied intervention continued after the trial upon request of the participating sites, but a decline in the quality of the ED registrations made some registrations unreliable. To explore the primary outcome after the trial ended, the number of patients originating from the ED as noted by the GPC receptionist was extracted from the GPC's software (Mediris 2.4) both for the study period and one year afterwards. Because the COVID-19 pandemic disrupted the Belgian healthcare system mainly during two waves, the months April, November, and December 2020 were excluded [23].

## Ethics

Ethical clearance waiving individual informed consent was obtained from the ethics committee of Antwerp University Hospital (reference 18/37/410) and from the local ethics committee of AZ Monica Deurne (reference 367).

## Monitoring

One and six months after the start of the trial, the research team presented interim results to the working group that prepared the study. All staff members and the hospital's ombudsperson were asked to report all serious adverse events possibly related to the study.

## Patient and public involvement

A lay person volunteering at the ED of a hospital not participating in this study was involved in the study design, she gave advice about the study protocol and tool. An advisory board with stakeholders from Eds, GPCs and universities gave advice about the study design, discussed the interim analysis, and gave feedback on the results.

## Analysis

The primary outcome expressed as a percentage will be reported with a 95% CI (which implicitly corresponds to testing the null hypothesis that this percentage is equal to zero). Bivariate logistic regression was used to calculate odds ratios (OR) of the dichotomous outcomes across multilevel categorical independent variables. Those variables found significant (alpha = 0.05) in the bivariate analysis were incorporated in the multivariate analysis. This multivariate analysis was started by creating three chi-square automatic interaction detection (CHAID) decision trees [24, 25]. Decision tree methodology is a data mining method used for developing prediction algorithms of a dichotomous target variable taking into account the interactions of the independent variables. The algorithm is non-parametric, can efficiently deal with large, complicated datasets, and can accept missing values. Decision trees based on Bonferroni-Holms corrected chi-square tests were constructed separately for the study tool parameters, patient characteristics and timing of presentation (weekend, time period, subjective crowding). K-fold cross validation was used to protect against overfitting. A final decision tree was fitted using the significant variables as they turned out in the three separate analyses. The significant variables in the decision trees were entered in a generalised mixed model considering that observations are nested in nurses, that is, with the nurse as a random intercept. To compare the primary outcome during the study period to the year 2020, an unpaired samples student's t-test was used. Details about the statistical analysis and data cleaning can be found in **S3 File**.

IBM SPSS version 26 was used for the CHAID analysis. The generalised linear model was created in Jamovi version 1.6 using the GAMLj module [26]. The epiR package in R version 4.0 was used to calculate predictive values with a 95% CI [27]. For all other analysis, JMP pro version 15 was used.

# Results

## Study population

In this study, 9964 patients were assessed for eligibility, of which 1806 patients were excluded mostly because of the lack of a national insurance number or because they were already triaged (Fig 2). The intervention group consisted of 6374 patients (78.1%) clustered in 37 weekends. The control group consisted of 1784 (21.9%) patients clustered in 10 weekends. On one bank holiday allocated to the intervention group, the study was unintentionally not conducted. The baseline characteristics of the patients in the intervention and in the control group were similar except for the subjective crowding at the ED (Table 1).

Out of the 22 nurses, five nurses worked significantly less during control weekends (lowest OR, 0.12 95%CI 0.04 to 0.38, p<0.01) and seven worked significantly more (highest OR, 3.63 95%CI 2.91 to 4.68).

## Primary outcome

For 80 out of the 6374 participants in the intervention group the assignment was unknown; almost half of them (n = 34) left without being seen, the others were seen at the ED. These 80 patients were excluded from the following analysis. Out of the remaining patients, 838/6294

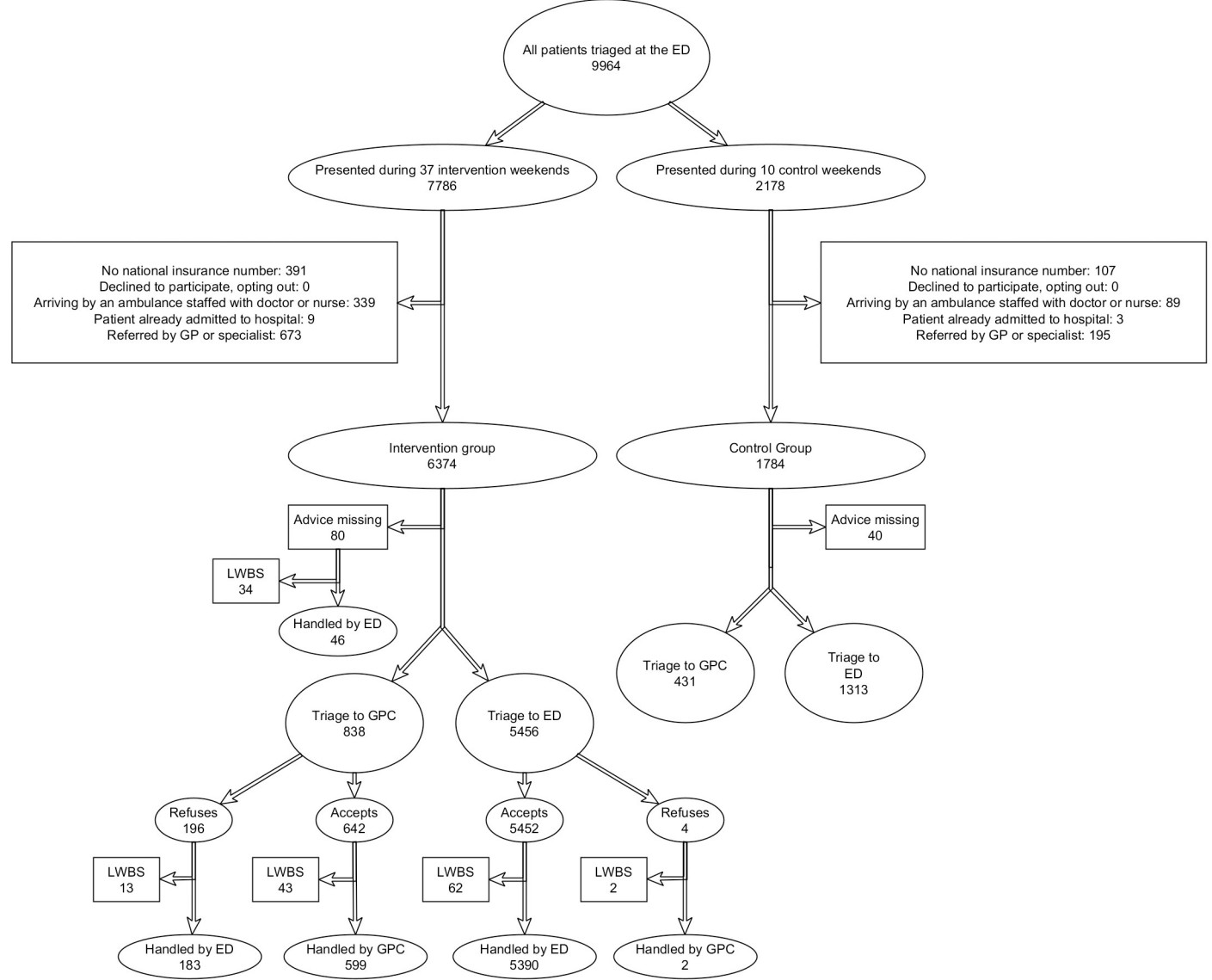

**Fig 2. Patient flow through the study (CONSORT flowchart).** ED: Emergency Department. GPC: General Practice Cooperative. GP: General Practitioner. LWBS: Left Without Being Seen.

(13.3%, 95% CI 12.5 to 14.2) were assigned to the GPC, 196/838 (23.4%, 95%CI 20.6 to 26.4) refused this assignment, 43/642 (6.7% 95%CI 5.0 to 8.9) accepted the assignment but left without being seen. The primary outcome was 599/6294 (9.5%, 95% CI 8.8 to 10.3). This primary outcome was 578/3098 (15.7% 95%CI 14.6–16.9) for patients within urgency category four and 21/59 (35.6% 95% CI 24.6–48.3) for patients within urgency category five. See Table 2 for the bivariate analysis of the primary outcome.

The most important determinants of the primary outcome were MTS urgency category (non-urgent versus standard, OR 2.96), MTS flowchart category ORL (otorhinolaryngology) complaints versus unwell adult OR 3.91), patient's age (above 74 years versus 40–54 years, OR 0.33), admission type (arrived by ambulance versus walk-in OR 0.05), subjective crowding at the ED (quiet versus normal OR 2.16), and nurse (nurse four versus nurse nine OR 3.56).

**Table 1. Baseline characteristics of participants.** Values are numbers (percentages).

| Characteristics | Intervention group (%) | Control group (%) | P-value |
|---|---|---|---|
| | (n = 6374) | (n = 1784) | |
| Mean age in years (standard deviation) | 38 (25) | 39 (24) | 0.11* |
| Sex | | | 0,95** |
| Women | 3149 (49) | 880 (49) | |
| Men | 3225 (51) | 904 (51) | |
| Residence | | | 0.14** |
| Nearby*** | 4481 (70) | 1217 (68) | |
| Others | 1873 (29) | 558 (31) | |
| Missing | 20 (0) | 9 (0) | |
| Socioeconomic Status | | | 0.18** |
| Low | 1642 (26) | 494 (28) | |
| Not low | 3716 (58) | 1027 (58) | |
| Missing | 1016 (16) | 263 (15) | |
| Manchester Triage System urgency category | | | 0.06** |
| One or two (max. waiting time ten minutes) | 413 (6) | 104 (6) | |
| Three (max. waiting time one hour) | 2146 (34) | 552 (31) | |
| Four (max. waiting time two hours) | 3726 (58) | 1097 (61) | |
| Five (max. waiting time four hours) | 89 (1) | 31 (2) | |
| Subjective crowding at the ED | | | <0.01** |
| Quiet | 272 (4) | 58 (3) | |
| Normal | 2127 (33) | 383 (21) | |
| Busy | 344 (5) | 92 (5) | |
| Missing | 3631 (57) | 1251 (70) | |
| Admission to the study hospital | 1018 (16) | 293 (16) | 0.65** |
| Mean number of included patients per weekend (standard deviation)) | 172 (39) | 178 (34) | 0.63* |

*P-value based on an unpaired samples student's t-test.

**P-value based on the Pearson's Chi-square test.

***Within the four communities covered by the GPC.

ED: Emergency Department.

CHAID analysis of the study tool parameters (**S1 Fig**) showed that the urgency category was primordial: none of the patients in the three highest urgency categories was seen at the GPC, patients within urgency category five were more likely to be diverted to the GPC as compared to urgency category four. Within urgency category four, the flowchart category became a determining factor. Abdominal complaints, ORL complaints, neurological complaints, respiratory complaints, children, unwell adult, and back neck pain led to the GPC in more than 30% of the cases, while limb problems, wounds, chest pain, eye problems, and mental complaints led to the GPC in 5.2% of the low urgency cases. CHAID analysis of the patient characteristics (**S2 Fig**) revealed admission type as the crucial factor, followed by socioeconomic status, residence, and age as the least determining variable. Finally, CHAID analysis of the timing of presentation (see **S3 Fig**) showed that during the day, the proportion of the primary outcome was lower when the subjective crowding was normal (7.5%) compared to quiet and busy ED (10.7%). A combined CHAID tree (Fig 3) demonstrates the pivotal role of the study tool components. Only in a selection of flowcharts the admission type and the time period played a significant role.

A comparison of the fixed effects generalised linear model (deviance 2200.6, df 53, AIC 2306.8, BIC 2636.7) with the generalised mixed model (deviance 2523.6, df 27, AIC 2577.6,

**Table 2. Logistic regression bivariate analysis of the primary outcome (all participants in the intervention weekends, excluding those with a missing assignment).** For categorical variables with more than four categories, the categories with the highest and lowest primary outcome are reported.

| Determinant | N | Mean primary outcome | DF | Category | Estimate | Wald Chi$^2$ | P-value | Odds ratio (95%CI) |
|---|---|---|---|---|---|---|---|---|
| Study tool parameters | | | | | | | | |
| MTS urgency category* | 3735 | 16.0% | 1 | 4: Standard | | | | 1 |
| | | | | 5: Non-urgent | 1.1 | 15.5 | <0.01 | 2.96 (1.73 to 5.08) |
| MTS flowchart category | 6238 | 9.4% | 14 | Unwell adult | | | | 1 |
| | | | | ORL Complaints | 1.4 | 51.9 | <0.01 | 3.91 (2.70 to 5.68) |
| | | | | Chest pain | -3.0 | 8.5 | <0.01 | 0.05 (0.01 to 0.38) |
| Patient characteristics | | | | | | | | |
| Age | 6294 | 9.5% | 5 | 0–7 years | 0.3 | 4.0 | <0.01 | 1.34 (1.00 to 1.79) |
| | | | | 8–24 years | 0.3 | 3.7 | 0.05 | 1.29 (1.00 to 1.68) |
| | | | | 25–39 years | 0.1 | 0.96 | 0.33 | 1.14 (0.88 to 1.48) |
| | | | | 40–54 | | | | 1 |
| | | | | 55–74 | -0.5 | 10.1 | <0.01 | 0.58 (0.41 to 0.81) |
| | | | | >74 | -1.1 | 22.6 | <0.01 | 0.33 (0.21 to 0.52) |
| Admission type | 6291 | 9.5% | 1 | Walk-in | | | | 1 |
| | | | | Arrived by ambulance | -2.32 | 62.6 | <0.01 | 0.10 (0.06 to 0.17) |
| Sex | 6294 | 9.5% | 1 | Female | | | | 1 |
| | | | | Male | -0.16 | 3.54 | 0.06 | 0.85 (0.72 to 1.01) |
| Residence | 6275 | 9.5% | 1 | Nearby | | | | 1 |
| | | | | Not living nearby | -0.47 | 20.4 | <0.01 | 0.63 (0.51 to 0.77) |
| Socioeconomic status | 5293 | 10.4% | 1 | Normal | | | | 1 |
| | | | | Low | | 14.7 | <0.01 | 1.42 (1.18 to 1.71) |
| Timing of presentation | | | | | | | | |
| Weekend | 6294 | 9.5% | 36 | 30/08/2019-02/09/2019 | | | | 1 |
| | | | | 11/10/2019-14/10/2019 | 0.67 | 4.12 | 0.04 | 1.94 (1.02 to 3.70) |
| | | | | 23/08/2019-26/08/2019 | -0.86 | 3.29 | 0.07 | 0.42 (0.17 to 1.07) |
| Time period | 6294 | 9.5% | 2 | Day | | | | 1 |
| | | | | Evening | -0.31 | 8.05 | <0.01 | 0.73 (0.59 to 0.91) |
| | | | | Night | 0.35 | 10.9 | <0.01 | 1.42 (1.15 to 1.76) |
| Subjective crowding at the ED | 2743 | 8.6% | 2 | Normal | | | | 1 |
| | | | | Quiet | 0.76 | 16.6 | <0.01 | 2.16 (1.49 to 3.12) |
| | | | | Busy | 0.25 | 1.6 | 0.21 | 1.29 (0.87 to 1.91) |
| Nurse | | | | | | | | |
| Nurse | 5967 | 9.9% | 21 | Nurse 9 | | | | 1 |
| | | | | Nurse 4 | 1.27 | 25.4 | <0.01 | 3.56 (2.17 to 5.83) |
| | | | | Nurse 20 | -1.06 | 2.02 | 1 | 0.35 (0.08 to 1.49) |

DF: degrees of freedom.

MTS: Manchester Triage System.

ED: Emergency Department.

ORL: Otorhinolaryngology.

*: Only for urgency categories four and five because the primary outcome was zero in the other category.

BIC 2745.7) led to rejection of the nurse's random effect in favour of the fixed effects model. In the generalised linear model (for patients within the urgency categories four and five only, N = 3735, mean primary outcome 16%), all variables had a significant effect on the primary outcome (pseudo R-squared .22), with the largest contribution by the MTS flowchart category (S3 Table).

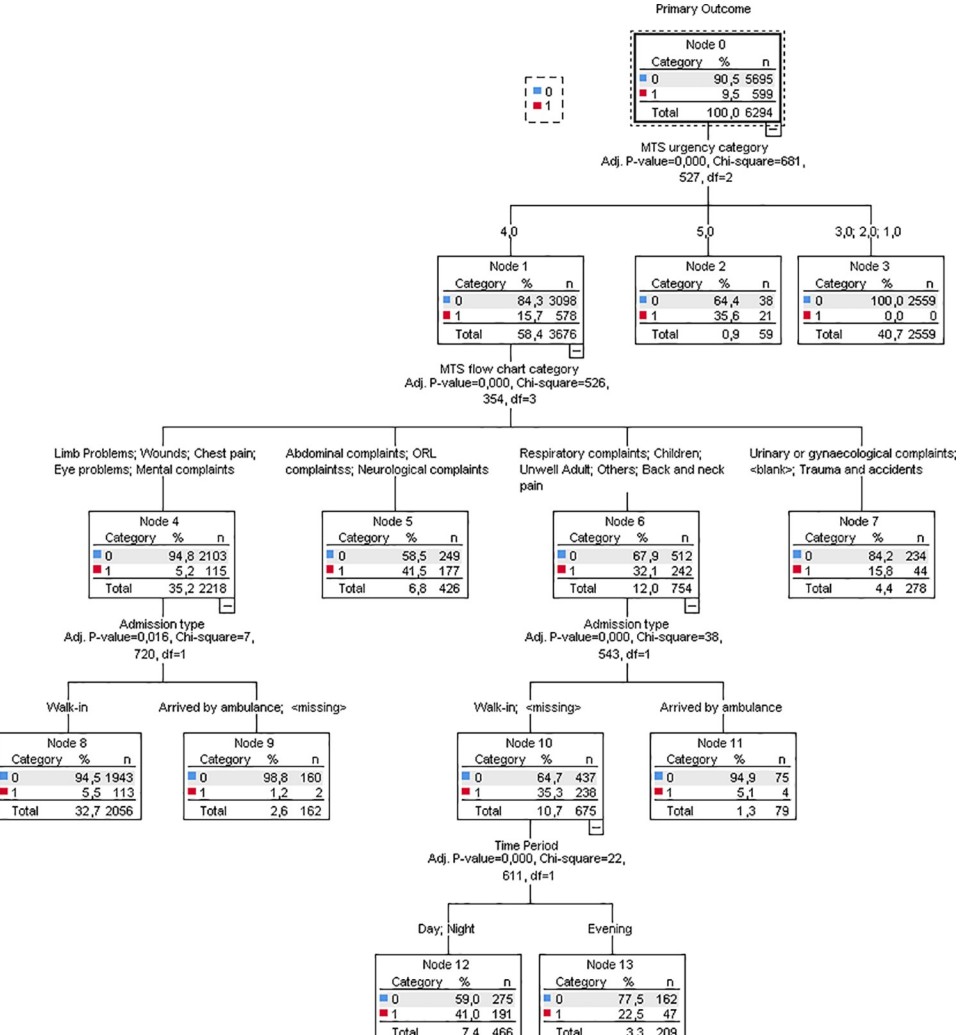

**Fig 3. Combined chi square aided interaction detection (CHAID) tree for the primary outcome.** ORL: Otorhinolaryngology. <blank>: no MTS flowchart was registered.

## Secondary outcome

During the intervention weekends, assignment to the GPC was recorded for 838 out of 6294 patients (13.3%, 95% CI 12.5 to 14.2); during control weekends this proportion was almost twice as high: 431/1744 (24.7%, 95% CI 22.7 to 26.8). Bivariate analysis (**S1 Table**) and CHAID analysis (**S4–S6 Figs**) of the secondary outcome gave results similar to the primary outcome, but the intervention became the most important determinant after the study tool components. The nurses overruled the automated eMTS assignment in 4.2% (95% CI 3.6–4.7) of the cases within urgency category four and five.

## Other outcomes

**Patients within the primary outcome referred back to the ED.** Out of the 599 patients within the primary outcome, the GPs referred 24 patients (4.0%, 95% CI 2.7 to 5.9) back to the ED (**S2 Table**). The proportion of these referrals was not significantly influenced by the triaging nurse, the patient's characteristics, or the weekend. Four out of the 19 patients with a

presentational flowchart category 'neurological complaints' were referred back to the ED, which was significantly higher compared to the reference category 'unwell adult' (OR 7.2 95% CI 1.2 to 43.2).

One potential serious adverse event was reported, a middle-aged male complaining of mild back pain presented at the ED at 2.00 PM. He was diverted to the GP who prescribed an analgesic and reassured him. Thirty minutes later a taxi brought the patient, who was having a cardiac arrest, back to the ED. During unsuccessful resuscitation a ruptured abdominal aneurysm was diagnosed. After assessing the records of this patient and an interview with the involved staff, the working group judged that the management of this patient would not have been different if he had been assigned to the ED.

**Admissions to the study hospital.**   The overall proportion of hospitalised patients was 1309/8038 (16.3% 95% CI 15.5 to 17.1). This proportion was 31/1236 (2.4% 95% CI 1.7 to 3.4) among patients with an assignment to the GPC. It was not influenced by the intervention (P = 0.56). Among the 599 patients in the primary outcome, three (0.5% 95% CI 0.2 to 1.5) were admitted to the hospital.

**Performance of the study tool to detect primary care patients.**   Patients without a known assignment to GPC or ED (n = 120) were excluded from this analysis. During intervention weekends, patients who refused the assignment (n = 196) were also excluded. For patients within the primary outcome (n = 599), the gold standard was referral by the GP. Patients referred back to the ED were considered "false positive" (n = 24), the others true positives (n = 575) leading to a positive predictive value of 0.96 (95% CI 0.94 to 0.97). For patients who accepted assignment to the ED (n = 5452), the gold standard was the opinion of the ED physician: false negative when these patients were eligible for primary care (n = 797) or true negative when these patients were not (n = 1196). The negative predictive value was 0.60 (95% CI 0.58 to 0.62).

During control weekends the gold standard was the opinion of the ED physician. The positive and negative predictive values were 0.84 (95% CI 0.78 to 0.90) and 0.56 (95% CI 0.51 to 0.60) respectively. The opinion of the ED physician was not registered at the ED in 4549/7196 (63%) of the patients. Eight out of the 24 physicians provided 94% of the values.

**Exploration of the primary outcome after the trial ended.**   The primary outcome during the intervention weekends calculated on the GPC data was 645, on average 16 patients (standard deviation 8.8) per weekend. In 2020, it remained 16 (standard deviation 6.4, p = 0.43).

## Discussion

In this trial, 838 (13%) out of the 6374 included patients in the intervention group were assigned to the GPC, of which 599 (71%) were seen at the GPC which rejects the null hypothesis of no diverted patients. Four percent of these patients were referred back to the ED. The primary outcome was mostly influenced by the study tool parameters: urgency category and chosen presentational flowchart. The remaining variability can be explained by factors related to the patient (mostly arriving by ambulance), the triaging nurse, and the timing of presentation. The secondary outcome was roughly influenced by the same determinants. During control weekends, this secondary outcome doubled. The positive and negative predictive values of the studied tool for detecting primary care patients were 0.96 and 0.60 during intervention weekends. The effectiveness remained unchanged after the trial, suggesting that the study induced longer-lasting structural changes in the triage and referral processes.

This study was the first cluster-randomised trial about diverting ED patients to primary care. Its strengths lie in the large number of included patients, its real live setting, and its long study period. This study has the universal limitations of a cluster-randomised trial, such as the

possibility of undetected imbalance among the study groups, interactions between individuals triaged after each other, and clustering of population characteristics on certain weekends. It was conducted in a single centre adapted to some local habits. The working group was not an independent data monitoring committee as they all worked in the studied services. The opinion of the ED physician about the assignment was well registered, but by a minority of the physicians, making the calculated predictive values prone to observer bias. There was an imbalance between intervention and control groups for the subjective crowding at the ED. This was probably due to a difference in motivation to register this parameter rather than due to an actual difference. An important difference between the studied tool and the original MTS is the discriminator GP Risk which allows a subjective judgment of the triaging nurse, this might reduce generalisability although it resembles everyday practice where nurses use their gut feelings [28]. The studied nurses were trained in the use of the MTS but whether or not the MTS was used correctly during the study period was not assessed. Finally, this study was only conducted during weekends and therefore the performance of the intervention during office hours or weeknights remains unknown.

A large study found a 22% increase of the proportion of patients attending the GP with a close collaboration with the ED as compared to the usual care setting, while another study found a decline in the number of patients treated at the ED by 20% after the introduction of a nearby GPC [29, 30] These results are similar to the secondary, but not the primary, outcome of the current study, so the effectiveness of the studied tool was rather low. Whether or not it is desirable to increase this primary outcome depends on whether the perspective is from the third-party payer, the patient, the service, or the healthcare professionals. This question will be answered in the upcoming process and financial analysis of the current study. Diverting 10% of the included patients to primary care reduced the workload at the ED, but the current study does not allow quantification of this impact. It might influence patient and staff satisfaction; a qualitative study about this aspect will be reported in the near future. The intervention might influence the long-term health seeking behaviour of patients as it is known that patients who were given the opportunity to be treated at a primary care clinic instead of an ED have increased future primary care follow-up compared with standard ED referral practices [31]. The higher primary outcome when the subjective crowding at the ED is quiet indicates that the ability of the studied tool to reduce crowding at the ED might be lower when it is needed the most.

The referral rate of patients in the primary outcome to the ED (4%) was similar to the referral rate of the studied GPC for untriaged patients (6%) and in general practice OOH services in the United Kingdom (8.1%) [32]. A retrospective study in which all self-referred, low-urgency patients were diverted to the GPC, found a referral rate back to the ED of 20% [33] The lower referral rate in the current study is probably due to the design of the studied tool. The power of the analysis of the patients referred back to the ED was limited due to their small number. The participating GPs might have increased their threshold for referring study patients back to the ED as these patients already came from the ED. The very low admission rate, both for patients within the primary outcome as for patients with an assignment to the GPC is an indicator of safety. The positive predictive value for an assignment to the GPC of 0.96 in the intervention and 0.84 in the control group is another indicator of safety. Because the studied tool seems safe for patients refusing an assignment to the GPC, it might be interesting to study the possibility to oblige patients to follow an assignment to the GPC. Long-term multicentre studies are necessary to confirm these safety findings in larger populations. These studies should also focus on commonly missed diagnoses such as myocardial infarction and pulmonary embolism as these can be disguised by unspecific symptoms. The MTS has been

found an acceptable tool for prioritising patients with symptoms of these diseases, but the current study does not allow extrapolation of these findings to the eMTS [34, 35].

Unfortunately, one patient presenting with abdominal pain diverted to the GPC deceased due to a ruptured abdominal aneurysm. It is impossible to draw conclusions based on this sole case, but it is an important warning for further research and implementation: a robust follow-up system for incidents related to triage is necessary. Previous research proved that the MTS is safe and does not underestimate the severity of the patients presenting with abdominal pain but within this research patients were not diverted to the GPC [36].

During intervention weekends, the number of patients left without being seen was higher in the group assigned to the GPC than in the group assigned to the ED. The authors received some anecdotal information about patients attending their own GP after the weekend but were not able to study what happened to all of these patients.

Most of the risk factors for the primary outcome (age, presentational flowchart, and timing of presentation) found in the multivariate analysis cannot be influenced by policy so the tool itself should be the focus for improvement. This seems feasible as the nurses followed the studied tool in 96% of the cases. The presentational flowchart 'limb problems' has the greatest potential as the nurses indicated a risk for referral to the GP in 1322/1803 (73%) low urgency cases mostly because they thought the patient needed radiology or sutures. In contrast to walk-in patients, those arriving with an ambulance had telephone triage before arriving at the ED. The low proportion of the primary and secondary outcome in these patients implies that is not useful to use studied tool afterwards. The differences in the primary outcome among nurses should be studied further and can be addressed by training. The higher primary outcome during the night demands further study: is it related to patient or nurse factors?

The secondary outcome was much higher in the control group than in the intervention group. It is probably easier for a nurse to write down a theoretical assignment to the GPC compared to discussing it with the patient. Qualitative and quantitative follow-up studies about this aspect will be reported soon. Training of the nurses might improve their ability to engage with patients to discuss the proper place of treatment.

## Conclusion

In this randomised trial about triaging patients to primary care, ED nurses using a new tool safely diverted 9.5% of the included ED patients to the GPC. Young patients arriving without an ambulance with a typical primary care presentation were more often assigned to the GPC. These results remained stable after the end of the trial. These results prove it is useful to implement triage using the eMTS but further multicentre studies with a focus on increasing the proportion of diverted low-risk patients and safety are needed.

## Supporting information

**S1 Checklist.**
(DOCX)

**S1 Fig. Primary outcome—chi-square automatic interaction detection (CHAID) decision tree of the study tool parameters.** MTS: Manchester Triage System. ORL: Otorhinolaryngology.
(PNG)

**S2 Fig. Primary outcome—chi-square automatic interaction detection (CHAID) decision tree of the patient characteristics.**
(PNG)

**S3 Fig. Primary outcome—chi-square automatic interaction detection (CHAID) decision tree of the timing of presentation.**
(PNG)

**S4 Fig. Secondary outcome—chi-square automatic interaction detection (CHAID) decision tree of the study tool parameters.** ED: Emergency Department. GP: General Practice. MTS: Manchester Triage System. ORL: Otorhinolaryngology.
(PNG)

**S5 Fig. Secondary outcome—chi-square automatic interaction detection (CHAID) decision tree of the patient characteristics.** ED: Emergency Department. GP: General Practice.
(PNG)

**S6 Fig. Secondary outcome—chi-square automatic interaction detection (CHAID) decision tree of the timing of presentation.** ED: Emergency Department. GP: General Practice.
(PNG)

**S7 Fig. Secondary outcome—chi-square automatic interaction detection (CHAID) combined decision tree.** ED: Emergency Department. GP: General Practice. MTS: Manchester Triage System. ORL: Otorhinolaryngology.
(PNG)

**S1 Table. Bivariate analysis of the secondary outcome (all participants excluding those with a missing triage advice).** For categorical variables with more than four categories, the categories with the highest and lowest secondary outcome are reported. DF: degrees of freedom. MTS: Manchester Triage System. ED: Emergency Department. ORL: Otorhinolaryngology. *: Only for urgency categories four and five because the primary outcome was zero in the other categories.
(DOCX)

**S2 Table. Characteristics of patients referred back to the ED after triage to the GPC.** ED: Emergency Department. GP: General Practitioner. MTS: Manchester Triage System.
(DOCX)

**S3 Table. Generalised Mixed Model for the primary outcome.** MTS: Manchester Triage System. Df: degrees of freedom.
(DOCX)

**S4 Table. Presentational flowchart categories.** ORL: Otorhinolaryngology.
(DOCX)

**S1 File. Research protocol.** Original research protocol.
(DOCX)

**S2 File. Minor changes to the study protocol.** Overview of the minor changes made to the study protocol after trial registration.
(DOCX)

**S3 File. Statistical analysis plan.**
(DOCX)

## Acknowledgments

The authors would like to thank Carolyn Daher for proofreading the manuscript, all participating healthcare providers at the emergency department of AZ Monica Deurne, the staff of

the general practice cooperative Antwerp East and especially Sander Naeyaert, Ragna Verlent, Joo-Ree Melis, Marc Timmermans, Arnoud Bonemeyer, Edwin Vanbeveren, Lotte Fivez and Guido Michielsen for their extensive help and advice.

## Author Contributions

**Conceptualization:** Stefan Morreel, Hilde Philips, Diana De Graeve, Koenraad G. Monsieurs, Jarl K. Kampen, Jasmine Meysman, Eva Lefevre, Veronique Verhoeven.

**Data curation:** Stefan Morreel, Jarl K. Kampen.

**Formal analysis:** Stefan Morreel, Jarl K. Kampen, Eva Lefevre.

**Funding acquisition:** Stefan Morreel, Hilde Philips, Diana De Graeve, Koenraad G. Monsieurs, Jarl K. Kampen, Veronique Verhoeven.

**Investigation:** Stefan Morreel, Hilde Philips, Diana De Graeve, Koenraad G. Monsieurs, Jarl K. Kampen, Jasmine Meysman, Eva Lefevre, Veronique Verhoeven.

**Methodology:** Stefan Morreel, Hilde Philips, Diana De Graeve, Koenraad G. Monsieurs, Jarl K. Kampen, Jasmine Meysman, Eva Lefevre, Veronique Verhoeven.

**Project administration:** Stefan Morreel, Eva Lefevre.

**Resources:** Stefan Morreel.

**Software:** Stefan Morreel, Jarl K. Kampen.

**Supervision:** Stefan Morreel, Hilde Philips, Diana De Graeve, Koenraad G. Monsieurs, Veronique Verhoeven.

**Validation:** Stefan Morreel, Hilde Philips, Diana De Graeve, Koenraad G. Monsieurs, Jarl K. Kampen, Jasmine Meysman, Veronique Verhoeven.

**Visualization:** Stefan Morreel.

**Writing – original draft:** Stefan Morreel.

**Writing – review & editing:** Stefan Morreel, Hilde Philips, Diana De Graeve, Koenraad G. Monsieurs, Jasmine Meysman, Veronique Verhoeven.

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
