## [Decision Letter · Decision Letter 0]

27 Aug 2021

PONE-D-21-25450

Triaging and Referring In Adjacent General and Emergency Departments (the TRIAGE trial): a cluster randomised controlled trial

PLOS ONE

Dear Dr. Morreel,

Thank you for submitting your manuscript to PLOS ONE. After careful consideration, we feel that it has merit but does not fully meet PLOS ONE’s publication criteria as it currently stands. Therefore, we invite you to submit a revised version of the manuscript that addresses the points raised during the review process.

Having intensively reviewed your revised draft, our external reviewers differed with their final recommendations, at least to some extent. Thus, I have double checked your revised version, to come to a more balanced decision (see R #1). All in all, our  identified shortcomings are considered reasonable with regard to both PLOS ONE’s quality standards and our readership's expectations. Therefore, we invite you to submit a carefully revised version of the manuscript that addresses EACH AND EVERY point raised during the current review process. Please note that a further non-convincing revision (not considered acceptable with regard to language, content, reviewers' constructive criticism, generalizable conclusions, and/or Authors' Guidelines) must lead to outright reject. 

We look forward to receiving your revised manuscript.

Kind regards,

Andrej M Kielbassa

Academic Editor

PLOS ONE

Journal Requirements:

Reviewers' comments:

Reviewer's Responses to Questions

**Comments to the Author**

1. Is the manuscript technically sound, and do the data support the conclusions?

Reviewer #1: No

Reviewer #2: Yes

Reviewer #3: Yes

2. Has the statistical analysis been performed appropriately and rigorously? 

Reviewer #1: Yes

Reviewer #2: Yes

Reviewer #3: Yes

3. Have the authors made all data underlying the findings in their manuscript fully available?

Reviewer #1: Yes

Reviewer #2: Yes

Reviewer #3: No

4. Is the manuscript presented in an intelligible fashion and written in standard English?

Reviewer #1: Yes

Reviewer #2: Yes

Reviewer #3: Yes

5. Review Comments to the Author

Reviewer #1: Abstract

- Maximum word count is 300. Please see Guidelines for Authors, and shorten accordingly.

- "(...) nurse-led triage using a new tool assigning patients (...)" - meaning remains unclear, please add information.

- "(...) in the control group 431/1744 (24.7%, 95% CI 22.7 to 26.8)." Please provide complete sentences.

- What is meant when referring to "MTS presentational flowchart"? Please clarify.

- "ED nurses using a new tool safely diverted 9.5% of the included patients to primary care." This does not refer to your aims/objectives. Please revise carefully.

Intro

- Again, please note that your draft does not follow the Authors' Guidelines. Revise referencing, "(...) in a primary care setting.[1-4]" must read "(...) in a primary care setting [1-4]." Revise thoroughly.

- "Three small non randomised trials about the MTS and diversion to primary care revealed promising results [14-16]." Rationale would seem unclear. Please elaborate both aims and objectives more clearly.

- What about your null hypothesis? Remember that H0 must be deducible from the foregoing thoughts.

Meths

- Again, please note that your draft does not follow the Authors' Guidelines. Revise for "Material and methods".

- Would seem complete.

Results

- Convincingly elaborated.

Disc

- Please stick to H0 when starting this section.

- "The referral rate of patients in the primary outcome to the ED (4%) was similar to the referral rate of the studied GPC for untriaged patients (6%) and in general practice OOH services in the United Kingdom (8.1%)." Reference missing.

Concl

- With your Conclusions, please stick exclusively to your revised aims. Do not simply repeat your results here. Instead, provide a reasonable and generalizable extension of your outcome.

Refs

- Please revise for uniform formatting. Compare "The American journal of emergency medicine. 2011;29(3):333-45.", "BMJ. 2020;368:m462. doi:10.1136/bmj.m462.", or "Journal of emergency management (Weston, Mass). 2016;14(5):349-64. doi: 10.5055/jem.2016.0300. PubMed PMID: 27873299.", and re-edit thoroughly.

In total, this is an interesting paper, easily intelligible, and should be worth following after revision.

- doi and PMID numbers will be sufficient.

Reviewer #2: Thank you for giving me the opportunity to review the article entitled: "Triaging and Referring In Adjacent General and Emergency Departments (the TRIAGE trial): a cluster randomised controlled trial". This is a very interesting study on a topic that is becoming more and more relevant and that is increasingly affecting EDs in countries with an open and public health care system.

The use of the GPC strategy can be interesting and the results of the study demonstrate this.

On the other hand, there are some points that might need clarification.

In the methods section it would be interesting to indicate which diagrams were not changed in this new eMTS version.

In the results section it is indicated that the majority of patients assigned to GPC were a blue code, it would be interesting to compare the proportions (total blue patients and percentage of patients who used GPC vs total green patients and percentage of patients who used GPC). This is because in the most well-known MTS studies it is known that the majority of patients are green and a small proportion blue, this could give an accurate idea of the impact of this strategy.

Also, in the results section it is indicated that most of the patients eligible for GPC were detected on weekends, why? Are there more weekend accesses, are there no other types of services presents? Are the proportions of codes equal between weekend and week (or are there more green and blue due to the absence of some services)?

On page 17 it is stated that one patient elected for the GPC strategy died. Regardless of the individual case, this may also be the case in other non-specific symptom presentations that result in a more serious underlying condition (epigastric pain in a diabetic patient that results in a acute myocardial infarction, mild abdominal pain that results in appendicitis). There are different reports on individual symptom presentations and the functionality of MTS, and some of these indicate that patients with green and blue codes have a subsequent severe and time-dependent pathology (DOI: 10.1016/j.ienj.2020.100842; DOI: 10.1177/1474515118777402; DOI: 10.1111/jocn.15635). This might be an interesting aspect to be addressed in the discussion.

In supplementary table S9-T2 it is reported that a patient with adult unwell chart presented with the diagnosis of headache and so on, was the correct application of MTS assessed? If not, it could result as a limitation of the study.

Another aspect that could be included as a limitation of the study is the fact that the GPC physician may have altered their performance in patients sent from triage. Could he have decreased the rates of patients sent back to the ED given this knowledge? This limitation is given by the type of study but still deserves to be mentioned in the limitations section.

Despite these minor concerns the article is very interesting and explores a useful strategy for the performance of the ED, the impact of this strategy could be revolutionary for EDs and triage. The article is well written and clear.

Reviewer #3: A cluster randomized control study was conducted which aimed to determine the effectiveness and safety of a tool diverting patients eligible for primary care from an ED to the adjacent general practitioner cooperative (GPC). The primary outcome was the proportion of patients assigned to and handled by the GPC during intervention weekends. Randomization took place only for the secondary outcome: determining the proportion of patients assigned to GPC during intervention and control weekends.

Minor revisions:

1- Table 1: State the statistical testing methods used to estimate the p-values.

2- Table 2: To improve clarity, modify the title to include “logistic regression bivariate analysis.”

6. PLOS authors have the option to publish the peer review history of their article (what does this mean?). If published, this will include your full peer review and any attached files.

Reviewer #1: No

Reviewer #2: **Yes: **Zaboli Arian, RN, Emergency Department, Hospital of Merano

Reviewer #3: No

---

## [Author Response · Author response to Decision Letter 0]

8 Sep 2021

We have rewritten the entire paper for this first revision according to the remarks and questions of the reviewers and the editors. The changes made are available in the "response to the reviewers". We are willing to revise it again if necessary.

---

## [Decision Letter · Decision Letter 1]

30 Sep 2021

Triaging and Referring In Adjacent General and Emergency Departments (the TRIAGE trial): a cluster randomised controlled trial

PONE-D-21-25450R1

Dear Dr. Morreel,

After having double checked your revised and re-submitted paper, I am pleased to inform you that your manuscript has been judged scientifically suitable for publication and will be formally accepted for publication once it meets all outstanding technical requirements.

Kind regards, congratulations and compliments, and stay healthy

Andrej M Kielbassa, Prof. Dr. med. dent. Dr. h. c.

Academic Editor

PLOS ONE

Additional Editor Comments (optional):

Reviewers' comments:

Reviewer's Responses to Questions

**Comments to the Author**

1. If the authors have adequately addressed your comments raised in a previous round of review and you feel that this manuscript is now acceptable for publication, you may indicate that here to bypass the “Comments to the Author” section, enter your conflict of interest statement in the “Confidential to Editor” section, and submit your "Accept" recommendation.

Reviewer #1: All comments have been addressed

Reviewer #2: All comments have been addressed

Reviewer #3: All comments have been addressed

2. Is the manuscript technically sound, and do the data support the conclusions?

Reviewer #1: Yes

Reviewer #2: Yes

Reviewer #3: (No Response)

3. Has the statistical analysis been performed appropriately and rigorously? 

Reviewer #1: Yes

Reviewer #2: Yes

Reviewer #3: (No Response)

4. Have the authors made all data underlying the findings in their manuscript fully available?

Reviewer #1: Yes

Reviewer #2: Yes

Reviewer #3: (No Response)

5. Is the manuscript presented in an intelligible fashion and written in standard English?

Reviewer #1: Yes

Reviewer #2: Yes

Reviewer #3: (No Response)

6. Review Comments to the Author

Reviewer #1: This revised and re-submitted draft would seem satisfying. Depending on the external reviewers' recommendations, it is considered ready to proceed.

Reviewer #2: Thank you for clarifying the points I highlighted in the previous review. I thank the authors for responding and editing their article according to the suggestions of the reviewers and the Editor.

I think this is a very interesting topic, it is well known that EDs around the world suffer from excessive influx for their organizations and these solutions and strategies are needed and it is extremely important to study them. That is why I think the article in its current form is appropriate for the journal.

Reviewer #3: (No Response)

7. PLOS authors have the option to publish the peer review history of their article (what does this mean?). If published, this will include your full peer review and any attached files.

Reviewer #1: No

Reviewer #2: No

Reviewer #3: No

---

## [Editor Report · Acceptance letter]

25 Oct 2021

PONE-D-21-25450R1 

Triaging and Referring In Adjacent General and Emergency Departments (the TRIAGE trial): a cluster randomised controlled trial 

Dear Dr. Morreel:

I'm pleased to inform you that your manuscript has been deemed suitable for publication in PLOS ONE. Congratulations! Your manuscript is now with our production department. 

Kind regards, 

on behalf of

Prof. Dr. med. dent. Dr. h. c. Andrej M Kielbassa 

Academic Editor

PLOS ONE